# Astrocyte-Inspired Hierarchical Routing for Enhanced Expert Specialization in Mixture-of-Experts Models

## Abstract

The Mixture-of-Experts (MoE) architecture is a leading paradigm for scaling, but cultivating genuine expert specialization is a persistent challenge, often hindered by load balancing. This paper introduces Astrocyte-Hierarchical Routing (AHR), a novel, bio-inspired mechanism that addresses this challenge. Drawing inspiration from astrocytes, AHR conditions local, token-level routing decisions on a global context signal. In our encoder-based implementation, this signal, derived from the [CLS] token, additively biases local routing decisions, promoting a developmental trajectory for expert functionality. We conduct experiments on a multi-class text classification task, comparing AHR against strong baselines. The results demonstrate that AHR achieves a statistically significant and substantial increase in final-layer expert specialization without incurring a discernible loss in task performance. Qualitative analysis further confirms that AHR fosters a transition from generalist experts in early layers to highly specialized experts in later layers. This work presents a new principle for MoE router design: a contextual, two-level approach. This successful validation in an encoder model serves as a proof-of-concept, opening the way for future work on scaling AHR and adapting its principle to other architectures.

## 1 Introduction

The current trajectory in deep learning is defined by an exponential increase in model scale, with state-of-the-art architectures now reaching trillions of parameters. While this growth has led to remarkable capabilities, it has also introduced substantial computational and energy constraints that challenge the sustainability of dense model training and deployment (Shazeer et al., 2017). The Mixture-of-Experts (MoE) architecture has emerged as a key solution, leveraging the principle of conditional computation. MoE fundamentally shifts the scaling paradigm: it dramatically increases a model's total parameter count by having the network activate only a small, sparse subset of its parameters (experts) for any given input, thereby boosting capacity without a commensurate rise in the required computational effort (Shazeer et al., 2017).

At the heart of the MoE paradigm is the concept of expert specialization. The architecture is designed to allow different experts—typically feed-forward networks (FFNs)—to become proficient in handling distinct subsets of data or computational patterns (Fedus et al., 2022). However, realizing this potential has proven to be a significant challenge. A critical issue in training MoE models is load imbalance, where the routing mechanism may favor a small subset of experts, leaving others undertrained (Shazeer et al., 2017). The standard solution is to introduce an auxiliary load-balancing loss, which encourages a more uniform distribution of tokens across experts (Shazeer et al., 2017). This necessity often creates a "specialization paradox": the auxiliary loss, in its effort to ensure uniform expert usage, can inadvertently inhibit the experts from developing distinct, specialized functions by promoting routing homogeneity (Liu et al., 2023). This problem is amplified when a micro-batch lacks diversity, as the load balancer may force the distribution of semantically similar tokens across numerous experts, thus directly undermining specialization (He et al., 2022).

To address this fundamental challenge, this work turns to the intricate computational principles of the brain. Traditional artificial neural networks are largely inspired by a neuron-centric view of computation. However, contemporary neuroscience reveals a more complex picture, highlighting the active role of glial cells,

particularly astrocytes, in information processing (Oschmann et al., 2018). The concept of the "tripartite synapse" posits that astrocytes are not mere support cells but are integral components of synaptic function. They integrate signals from thousands of surrounding synapses and, in turn, release gliotransmitters that modulate synaptic strength and neuronal activity (Oschmann et al., 2018). This ability to provide context-aware, localized regulation based on integrated network activity presents a powerful computational metaphor for a more sophisticated routing mechanism in MoE models. Astrocytes demonstrate that effective biological computation relies on mechanisms that are neither purely local nor naively global, but are instead contextual and hierarchical (Serra et al., 2022).

This paper introduces Astrocyte-Hierarchical Routing (AHR), a novel routing strategy that operationalizes this biological principle. AHR reframes the routing problem from a series of independent, token-level decisions to a contextual, two-level process. It conditions the routing decision for a token at a given layer on a global, sequence-level context vector (here derived from the [CLS] token). This hierarchical dependency—where a global signal biases local token routing—encourages the model to learn consistent processing pathways, allowing generalist experts in early layers to feed into progressively more specialized experts in later layers. This approach moves beyond simply selecting which experts to use at a given moment and instead focuses on how to orchestrate their activation across the depth of the network to cultivate distinct identities.

Through a series of controlled experiments, this study demonstrates that AHR successfully resolves the specialization paradox. The results show that AHR achieves task accuracy that is statistically indistinguishable from a highly-tuned softmax-gated baseline. Concurrently, it yields a dramatic and statistically significant increase in expert specialization in the final layer of the network, a critical stage for abstract feature processing. This work thus makes a twofold contribution: it presents a practical, high-performing routing algorithm that fosters genuine expert specialization, and it introduces a new design principle for MoE architectures based on contextual, global-plus-local routing.

## 2 Related Work

### 2.1 Mixture-of-Experts Architectures

The modern MoE layer, initially proposed as the Sparsely-Gated Mixture-of-Experts, represents a fundamental form of conditional computation designed to partition the computational load using multiple dedicated sub-networks, or "experts" (Shazeer et al., 2017). A typical MoE block consists of several expert FFNs and a router that dynamically selects which subset of experts processes a given token. The Switch Transformer greatly advanced this architecture by demonstrating that even a simple Top-1 selection strategy could efficiently scale models to trillions of parameters, dramatically accelerating the pre-training process (Fedus et al., 2022). Current best practices predominantly use a Top-K sparsely-gated approach to maintain efficiency. The router estimates the utility of each expert for an input token via a probability distribution (often Softmax) and selects the k experts with the highest scores (e.g., k=1 or k=2), although some strategies propose dynamically adjusting this number based on input difficulty (Huang et al., 2024). It is this conditional computation that enables MoE models to scale their parameter count into the trillions without a proportional increase in the required floating-point operations (FLOPs) per input token (Fedus et al., 2022).

A critical challenge in training these models is ensuring a balanced utilization of all experts. This was a problem the foundational MoE paper addressed by introducing an initial auxiliary load-balancing loss (Shazeer et al., 2017). This loss penalizes imbalances in both the number of tokens assigned to each expert and the routing weights themselves, encouraging a more uniform distribution. Recognizing the potential negative impact of this auxiliary loss, recent research has explored alternatives, such as auxiliary-loss-free balancing strategies (Wang et al., 2024) and system-level optimizations like those in DeepSpeed-MoE, which improve throughput and advance MoE inference and training at scale (Rajbhandari et al., 2022).

### 2.2 The Challenge of Expert Specialization

The goal of MoE is to achieve expert specialization: the emergent property where distinct experts become proficient in processing specific data subsets or computational functions (Fedus et al., 2022). This functional

differentiation is what justifies the increased capacity. Specialization is often quantified by measuring the variance in expert utilization across different input categories; high variance suggests selective activation for specific data types. The core problem, however, is that the standard load-balancing loss actively inhibits this differentiation by enforcing uniform routing. This led to OMoE, which attempts to resolve the specialization paradox by introducing an orthogonal optimizer to enforce representational diversity (Liu et al., 2023). Other approaches to encourage specialization include using a complementary orthogonality loss (Liu et al., 2023) or, as seen in Switch Transformers, simplifying the selection to Top-1 routing. This Top-1 strategy encourages the model to generate a larger pool of highly specialized experts, preventing any one expert from attempting to cover an excessively broad range of knowledge (Fedus et al., 2022). The method proposed in this paper, AHR, offers a novel, architecturally-driven alternative that fosters specialization by structuring the flow of information between layers, rather than through modifications to the loss function or expert design alone.

### 2.3 Advanced and Dynamic Routing Strategies

MoE research is moving toward dynamic, context-aware routing strategies that replace static, uniform mechanisms. The traditional Top-K approach is an instance of a "tokens choose experts" method, where each token independently selects its best match. In contrast, the "experts choose tokens" paradigm is represented by Expert Choice (EC) Routing. EC Routing inverts the process, enabling each expert to select the top-k tokens from the batch that it is best equipped to handle, which naturally addresses load imbalance and provides tokens with a variable number of experts (Zhou et al., 2022). Another research direction makes the number of activated experts (k) dynamic. For example, Ada-K routing operates on the principle that not all tokens are equally complex; it employs a reinforcement learning (PPO) framework to dynamically determine the optimal K for each token (Yue et al., 2024). Supporting this idea, confidence-based routing strategies also confirm the notion that "Harder tasks need more experts," allowing simpler inputs to be processed sparsely (Huang et al., 2024).

This pursuit of global context is shared by other recent SOTA methods. For instance, **GLIDER** (Li et al., 2024) enables contextual routing for PEFT expert modules (LoRAs) by using a semantic global router guided by LLM-generated instructions combined with a learned local router. Similarly, **THOR-MoE** (Liang et al., 2025) features a hierarchical design for language- and domain-specific routing and injects context information into routing decisions, validated on multilingual Neural Machine Translation (NMT) benchmarks. Other methods have focused on improving the core selection mechanism itself, such as **DSelect-k** (Hazimeh et al., 2021), which introduces a differentiable proxy for the non-differentiable Top-K selection for multi-task learning problems, or **PMoE** (Ren et al., 2021), which uses probabilistic sampling instead of deterministic selection to decompose policy learning in Deep Reinforcement Learning (DRL) applications.

AHR synthesizes and extends the "global context" idea from methods like GLIDER but implements it in a novel and far simpler way. Instead of relying on complex instruction-following modules or a task-level optimization framework, AHR uses a simple, fully-differentiable additive bias derived from the global representation (e.g. [CLS] token), achieving contextual routing without the associated training complexity.

### 2.4 Bio-Inspired Neural Networks and Astrocytic Computation

The history of artificial intelligence is rich with inspiration drawn from computational neuroscience. Recently, the focus has expanded beyond neurons to include glial cells, particularly astrocytes, which are now understood to be active participants in neural computation, as reviewed in works covering in silico models of neuron-astrocyte networks (Oschmann et al., 2018). Astrocytes are crucial for sensing neural activity and regulating synaptic plasticity and strength, playing a fundamental role in learning and memory, which has inspired computational models of neuron-astrocyte associative memory (Kozachkov et al., 2025). They integrate local synaptic activity with global neuromodulatory signals to control circuit function, ion homeostasis, and even sleep-wake cycles, for instance by promoting sleep via adenosine release (Haydon, 2017).

This growing biological understanding has inspired the creation of computational models incorporating astrocytic functions. Some work has focused on simulating the tripartite synapse to investigate memory and neural firing patterns (Oschmann et al., 2018). Other models, like AstroNet, employ the astrocyte as a metaphor for network optimization in computer vision problems, introducing an astrocyte-inspired unit

to govern network pruning and connection management (Han et al., 2023). It is critical to distinguish the present work from AstroNet. AstroNet operates by introducing a parallel "Astrocyte Network" responsible for learning the primary network's structure. It achieves efficiency by adaptively re-weighting or pruning the static connections of the main network, thus modifying the architecture over time (Han et al., 2023). In contrast, AHR applies the astrocyte metaphor to solve the dynamic routing problem specific to sparse MoE architectures. AHR does not alter the static weights of the experts; it governs the conditional computation pathways that are activated for each input on-the-fly. This fundamental difference in both the problem domain and the technical mechanism establishes the novelty of our contribution.

## 3 Astrocyte-Hierarchical Routing (AHR)

### 3.1 Conceptual Framework: From Biological Astrocytes to AI Routers

The design of Astrocyte-Hierarchical Routing is conceptually grounded in the integrative and modulatory function of biological astrocytes. A single astrocyte in the brain does not interact with just one synapse; it envelops thousands, allowing it to integrate information over a local network domain (Oschmann et al., 2018). Its activity is driven by two types of signals: local signals, in the form of neurotransmitters released from nearby active neurons, and more global signals, such as neuromodulators that convey the brain's overall state (Kozachkov et al., 2025). By integrating these signals, the astrocyte can release its own chemical messengers (gliotransmitters) to modulate the activity of the very circuits it is monitoring, creating a sophisticated feedback loop.

AHR translates this biological process into a computational mechanism for MoE routing. In our framework:

- **The Local Signal** is the input token's representation at a given MoE layer. This vector contains the immediate, local information needed for a routing decision.

- **The Integrated Contextual Signal** is a single vector that represents the global, sequence-level context, analogous to the integrated synaptic activity monitored by an astrocyte. In the encoder-based architecture of this study, we derive this signal from the `[CLS]` token's representation at the *current* layer.

- **Modulation** is performed by the AHR router, which uses this global [CLS] context to generate a bias that is applied to the routing logits of the current layer. This effectively modulates the expert selection process, biasing it based on the *overall meaning* of the sequence.

### 3.2 Baseline: Standard Top-K Softmax Gating

To provide a clear point of comparison, we first formalize the standard Top-K softmax gating mechanism used in our baseline models. Let $X \in \mathbb{R}^{S \times d}$ be the input sequence of $S$ token representations, each of dimension $d$, at a given MoE layer. The gating network, parameterized by a learnable weight matrix $W_r \in \mathbb{R}^{d \times N}$ where $N$ is the number of experts, computes a matrix of logits:

$$L_{\text{token}} = X \cdot W_r \quad (\text{where } L_{\text{token}} \in \mathbb{R}^{S \times N}) \tag{1}$$

For each token (i.e., each row $s$ of $L_{\text{token}}$), these logits are converted into a probability distribution $P_s \in \mathbb{R}^N$ over the experts using the softmax function:

$$P_s = \text{Softmax}(L_{\text{token},s}) \tag{2}$$

A function $\text{TopK}(P_s, k)$ returns the indices $\mathcal{T}_s$ of the $k$ largest values in $P_s$. The final gating weights $g_s(X) \in \mathbb{R}^N$ for token $s$ are computed by normalizing the probabilities of the selected experts and setting all others to zero (Huang et al., 2024):

$$g_{s,i}(X) = \begin{cases} \frac{P_{s,i}}{\sum_{j \in \mathcal{T}_s} P_{s,j}} & \text{if } i \in \mathcal{T}_s \\ 0 & \text{if } i \notin \mathcal{T}_s \end{cases} \tag{3}$$

The output of the MoE layer, $Y \in \mathbb{R}^{S \times d}$, is the weighted sum of the outputs from the selected expert networks $E_i(X_s)$ for each token:

$$Y_s = \sum_{i=1}^{N} g_{s,i}(X) E_i(X_s) \tag{4}$$

### 3.3 Method Formulation: Astrocyte-Hierarchical Routing (AHR)

AHR modifies the standard routing process by incorporating a global context signal. Let $X \in \mathbb{R}^{S \times d}$ be the input hidden states at layer $L$. First, the standard token-level logits are computed as in the baseline, using the layer's gating matrix $W_r \in \mathbb{R}^{d \times N}$:

$$L_{\text{token}} = X \cdot W_r \tag{5}$$

Next, we define the "astrocytic context vector" by extracting the [CLS] token's representation, which is the first token in the sequence: $x_{\text{cls}} = X_0 \in \mathbb{R}^d$. A new learnable parameter, the astrocytic projection matrix $W_a \in \mathbb{R}^{d \times N}$, transforms this context vector into a global bias term:

$$L_{\text{global}} = x_{\text{cls}} \cdot W_a \quad (\text{where } L_{\text{global}} \in \mathbb{R}^N) \tag{6}$$

This bias vector is then added to the standard token-level logits. This operation broadcasts $L_{\text{global}}$ across the sequence dimension $S$, applying the same global bias to every token's routing decision:

$$L_{\text{mod}} = L_{\text{token}} + L_{\text{global}} \tag{7}$$

The remainder of the routing process—applying the softmax function, selecting the Top-K indices, and normalizing the weights—proceeds as in the baseline, but is applied to these $L_{\text{mod}}$. The matrix $W_a$ is learned via backpropagation along with the other model parameters. It effectively learns to bias the expert selection for all tokens in the sequence based on the sequence's global semantic context, as captured by the [CLS] token.

### 3.4 Ablation: Astrocyte-Meanpool

To investigate the importance of using the [CLS] token as the context source and an additive bias as the modulation mechanism, an ablation study was conducted with a variant named `astrocyte-meanpool`. In this configuration, instead of using the [CLS] token, it computes a global context vector $x_{\text{mean}}$ by taking the mean of all other token representations in the sequence (i.e., $X_{1:S}$). This vector is then passed through a modulator (a linear layer followed by a Sigmoid function) to produce multiplicative weights $w_{\text{global}} \in \mathbb{R}^N$. These weights are then broadcast and multiplied with the token-level logits: $L_{\text{mod}} = L_{\text{token}} * w_{\text{global}}$. This ablation tests whether an additive bias (AHR) is more effective than multiplicative gating, and whether the [CLS] token is a better context signal than the mean of all other tokens.

### 3.5 Hypothesized Impact on Specialization

The hierarchical structure of AHR (global context modulating local decisions) is hypothesized to be a key driver of enhanced expert specialization. By providing all tokens in a sequence with a shared bias based on the sequence's topic, AHR encourages the model to learn consistent routing pathways for semantically

similar inputs. An expert in a deeper layer, say layer *L*, will be trained on a more curated set of inputs, as it will be preferentially selected for tokens belonging to sequences that generated a specific global bias. This progressive curation of the input distribution for each expert is the core mechanism that we posit drives specialization. This effect is expected to be cumulative, leading to the most pronounced functional differentiation in the final layers of the network, where representations are most abstract.

## 4 Experiments

### 4.1 Task and Dataset

The efficacy of the proposed AHR method was evaluated on a multi-class text classification task. The experiments utilized the standard AG News benchmark dataset, which comprises four distinct and balanced categories: World, Sports, Business, and Sci/Tech. Standard text preprocessing was applied, including tokenization using a pre-trained BERT word-piece tokenizer, conversion to numerical IDs, and padding or truncating sequences to a fixed length of 128 tokens.

### 4.2 Model Configurations

All experiments were based on a small transformer encoder architecture (`google/bert_uncased_L-4_H-512_A-8`). MoE layers, each containing $E = 8$ experts, were used to replace the standard FFN sub-blocks in all 4 transformer layers. The following model configurations were systematically compared:

- `dense`: A standard dense transformer model serving as a non-MoE performance baseline.

- `dense-random-ffn`: A dense model where the FFNs were randomly re-initialized, serving as a fair comparison for the MoE models which start with randomly-initialized routers.

- `softmax-moe`: A standard MoE model using Top-K (K=2) softmax gating with a load balancing (CV) loss.

- `softmax-moe-energy`: A `softmax-moe` model with an additional L2 "energy" loss on router probabilities.

- `astrocyte`: A variant using the [CLS] token for multiplicative (Sigmoid) modulation.

- `astrocyte-no-energy`: The `astrocyte` model without the energy loss.

- `astrocyte-unleashed`: An `astrocyte` model with no load balancing loss and a high energy loss.

- `astrocyte-hierarchical`: The primary proposed model (AHR) implementing the additive [CLS] token bias as described in Section 3.3.

- `astrocyte-meanpool`: The ablation model described in Section 3.4, using mean-pooled context for multiplicative modulation.

### 4.3 Evaluation Protocol

To ensure the reliability and reproducibility of the results, all experiments were conducted with $N = 5$ different random seeds. The reported results represent the mean and standard error across these five runs. Models were trained for 6 epochs using the AdamW optimizer with a linear learning rate warmup. The following metrics were used for evaluation:

- **Mean Evaluation Accuracy**: The primary metric for task performance, calculated as the percentage of correctly classified examples on a held-out test set.

- **Expert Specialization Score**: This metric was designed to quantify the degree of functional specialization among experts. For each MoE layer, we first compute an expert utilization matrix $U \in \mathbb{R}^{C \times N}$, where $C$ is the number of text categories (4) and $N$ is the number of experts (8). Each entry $U_{ij}$ represents the percentage of tokens from topic $i$ that are routed to expert $j$. The specialization score for that layer is then calculated as the average of the standard deviations of the columns of $U$ (i.e., $\text{mean}(\text{std}(U, \text{axis} = 0))$). A higher score indicates greater variance in an expert's utilization across different topics, signifying that the expert is more selectively activated for specific types of content.

Statistical significance between model configurations was assessed using an independent two-sample t-test. The results are reported with standard p-value thresholds: * $p{<}0.05$, ** $p{<}0.01$, and *** $p{<}0.001$.

## 5 Results and Analysis

### 5.1 Task Performance: Accuracy Comparison

The primary task performance results are presented in Figure 1. The central finding is that the proposed `astrocyte-hierarchical` model (AHR) achieves a mean evaluation accuracy of 0.9423. This performance is statistically indistinguishable from the strongest dense baseline (0.9431, $p{=}0.298$) and the strongest softmax-gated MoE baseline (`softmax-E8-LB0.01-EN0.01`, 0.9415, $p{=}0.160$). This result is of critical importance. It demonstrates that the introduction of the AHR mechanism, which is explicitly designed to foster a different learning dynamic and promote specialization, does not come at the expense of the model's primary objective: task accuracy. AHR successfully navigates the trade-off between encouraging specialization and maintaining high performance. Furthermore, the results show that several MoE configurations significantly outperform the `dense-random-ffn` baseline ($p{<}0.001$), validating the general effectiveness of the MoE approach for this task.

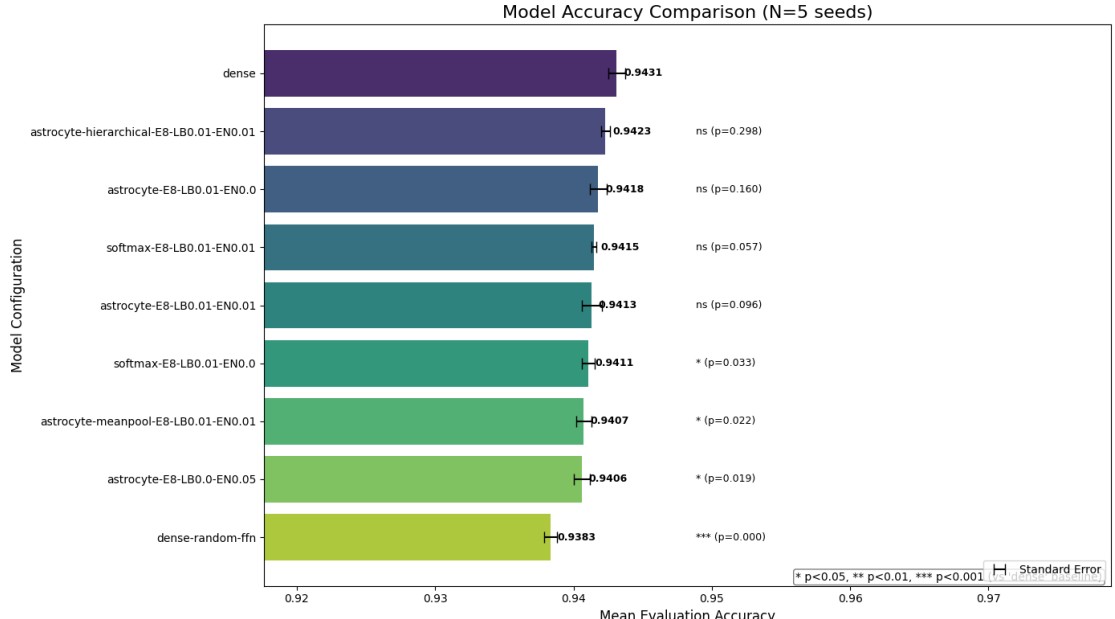

Figure 1: Model Accuracy Comparison (N=5 seeds). The proposed `astrocyte-hierarchical` model (0.9423) achieves accuracy statistically indistinguishable from the strongest dense (0.9431) and softmax (0.9415) baselines, demonstrating no trade-off in task performance. P-values are relative to the `dense` baseline, except where noted.

## 5.2 Quantitative Analysis of Expert Specialization

The quantitative impact of AHR on expert specialization is shown in Figures 2 and 3, which compare the specialization scores in the first and last MoE layers of the network, respectively.

In the first layer (Figure 2), the `astrocyte-hierarchical` model exhibits a mean specialization score of approximately 0.0139. This is notably lower than the scores achieved by the softmax baselines, such as `softmax-E8-LB0.01-EN0.01` (0.0164). This indicates that in the early stages of processing, AHR does not enforce—and may even discourage—strong specialization, allowing experts to function as generalists.

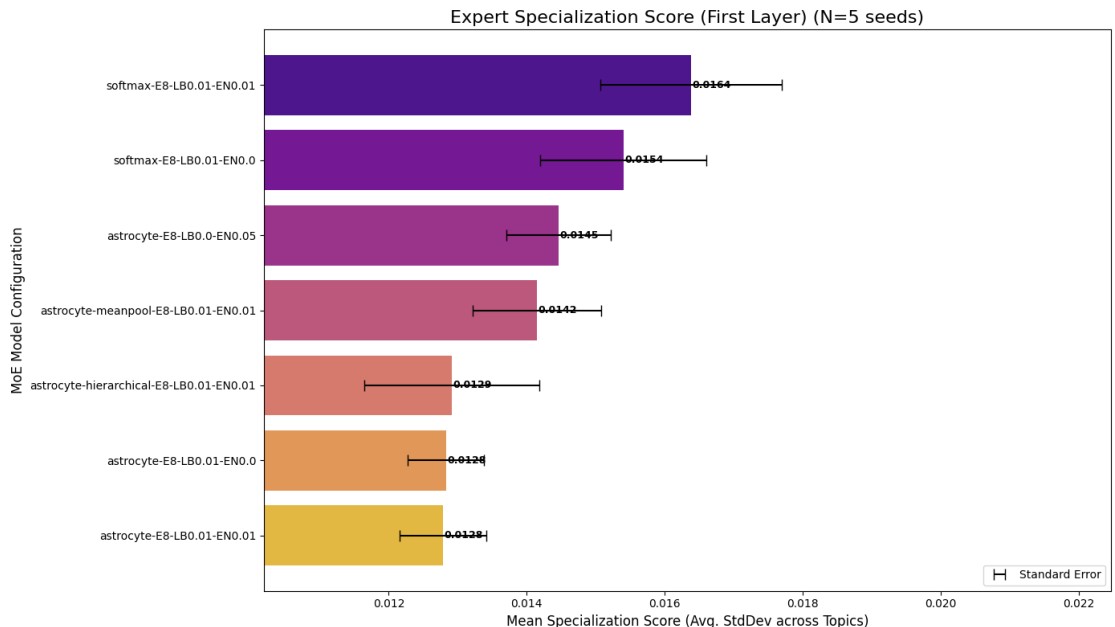

Figure 2: Expert Specialization Score (First Layer) (N=5 seeds). In the first layer, the AHR model (`astrocyte-hierarchical-E8-LB0.01-EN0.01`, 0.0139) shows lower specialization than softmax baselines (e.g., 0.0164), suggesting experts function as generalists.

The results for the last layer (Figure 3) reveal a starkly different and compelling story. Here, the `astrocyte-hierarchical` model achieves a mean specialization score of 0.1088. This is dramatically and statistically significantly higher than all other configurations. For comparison, the best-performing softmax baseline (`softmax-E8-LB0.01-EN0.0`) achieves a score of only 0.0598. The AHR model's specialization score is nearly double that of its strongest competitors ($p<0.001$ vs. all other MoE types). This sharp contrast between the first and last layers provides the core evidence for our hypothesis. AHR facilitates a hierarchical learning process where general-purpose experts process low-level features in the early layers, and this information is then routed to highly specialized experts that handle high-level, abstract features in the later layers.

## 5.3 Qualitative Analysis of Routing Behavior

A qualitative analysis of the expert utilization heatmaps provides a clear visual confirmation of the quantitative specialization scores. Figure 4 displays the routing patterns for a standard Softmax model, while Figure 5 shows the patterns for the Astrocyte-Hierarchical model.

The Softmax model (Figure 4) exhibits a relatively diffuse routing pattern across all layers. In Layer 0, experts E5 and E6 are used broadly across topics. Even in the final layer (Layer 3), while some specialization emerges (e.g., expert E1 is biased towards Sports), many experts remain generalists. For instance, expert E7 is utilized substantially for all four topics, and expert E0 is activated for World, Business, and Sci/Tech.

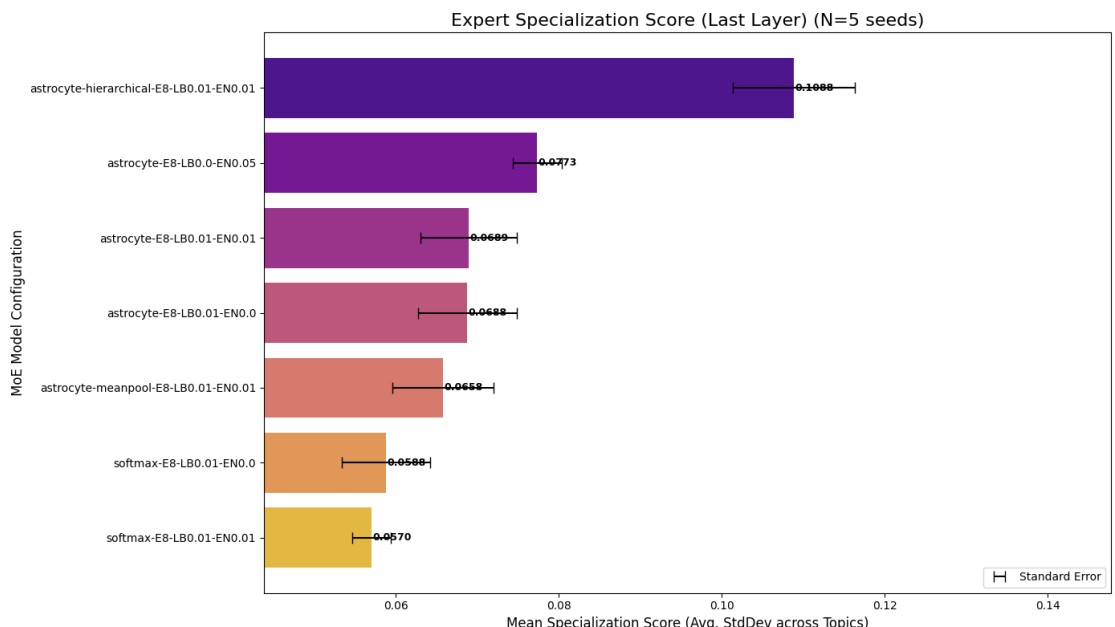

Figure 3: Expert Specialization Score (Last Layer) (N=5 seeds). In the last layer, the AHR model (`astrocyte-hierarchical-E8-LB0.01-EN0.01`) achieves a mean specialization score of 0.1088, dramatically and statistically significantly higher than all other configurations, including the best softmax baseline (0.0598).

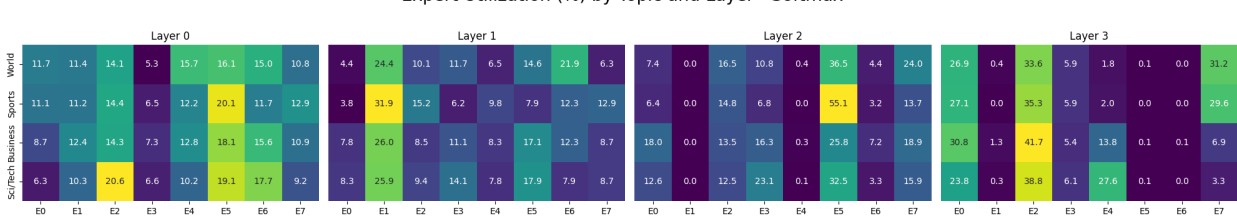

Figure 4: Expert Utilization (%) by Topic and Layer - Softmax. The baseline Softmax model shows a diffuse routing pattern, with many experts remaining generalists even in the final layer (Layer 3) (e.g., E7 is used for all topics).

The Astrocyte-Hierarchical model (Figure 5) presents a fundamentally different dynamic. In Layer 0, the routing is intentionally generalist; experts E0, E1, E6, and E7 are activated with high frequency for all topics, serving as a common processing front-end. However, as information propagates through the network, a highly structured and specialized pattern emerges. By Layer 3, the expert roles have become sharply defined. Expert E5 is almost exclusively dedicated to the "Sports" category, with a 59.6% utilization rate. Expert E0 has become the primary "Sci/Tech" specialist, handling 52.7% of tokens from that category. Expert E7, while handling a mix, is strongly biased towards "Business" and "World," and expert E4 has become the dedicated "Sports" backup. These heatmaps intuitively illustrate how AHR achieves its high final-layer specialization score. It does not force specialization from the outset. Instead, it guides a developmental process, starting with generalists and progressively refining expert roles until they become functionally distinct modules for high-level concepts.

Expert Utilization (%) by Topic and Layer - Astrocyte-Hierarchical

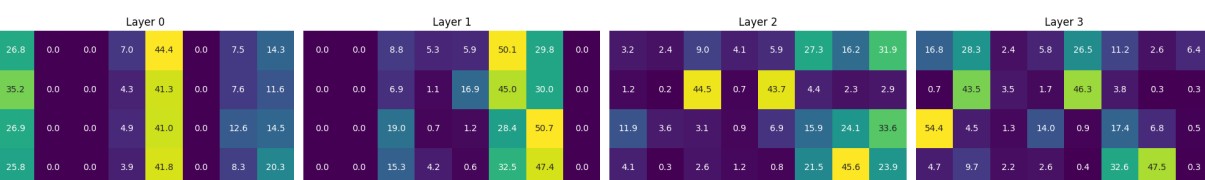

Figure 5: Expert Utilization (%) by Topic and Layer - Astrocyte-Hierarchical. The AHR model shows generalist experts in Layer 0 (e.g., E0, E1, E6, E7) that transition to highly specialized experts by Layer 3 (e.g., E5 for "Sports", E0 for "Sci/Tech").

Table 1: Comparison of Model Performance and Specialization. This table consolidates the key metrics, showing AHR achieves superior last-layer specialization while maintaining competitive accuracy.

| Model Configuration | Mean Accuracy (± SE) | p-value vs. dense | Specialization (First Layer) | Specialization (Last Layer) |
|---|---|---|---|---|
| dense | $0.9431 \pm 0.0005$ | - | N/A | N/A |
| astrocyte-hierarchical-E8-LB0.01-EN0.01 | $0.9423 \pm 0.0007$ | ns ($p$=0.298) | 0.0139 | 0.1088 |
| softmax-E8-LB0.01-EN0.01 | $0.9415 \pm 0.0009$ | ns ($p$=0.057) | 0.0164 | 0.0570 |
| softmax-E8-LB0.01-EN0.0 | $0.9411 \pm 0.0008$ | * ($p$=0.033) | 0.0154 | 0.0598 |
| astrocyte-meanpool-E8-LB0.01-EN0.01 | $0.9407 \pm 0.0008$ | * ($p$=0.022) | 0.0143 | 0.0658 |

## 5.4 Consolidated Findings

To provide a comprehensive overview of the key trade-offs and results, the performance and specialization metrics for the most salient model configurations are consolidated in Table 1. This table allows for a direct comparison, highlighting that the astrocyte-hierarchical model achieves its superior specialization in the last layer while maintaining a competitive level of accuracy.

# 6 Discussion

## 6.1 Synthesis of Findings: A Developmental Trajectory for Experts

The collective results paint a clear picture of the unique learning dynamic induced by Astrocyte-Hierarchical Routing. The key finding is not merely that AHR increases specialization, but rather the specific nature of this increase: low specialization in early layers followed by high specialization in later layers. This suggests that AHR imposes a valuable inductive bias that encourages a "developmental trajectory" for expert function. The initial MoE layers act as a shared, general-purpose feature extraction base, where a broad set of experts process incoming tokens. This foundation is then leveraged by the final layers, where experts can afford to become highly specialized in abstract, high-level concepts, as they are conditioned on a pre-processed and structured input stream. This contrasts sharply with standard MoE training, which implicitly attempts to force specialization from the very first layer—a potentially more difficult learning problem, as experts must simultaneously learn low-level feature extraction and high-level semantic differentiation.

## 6.2 Implications and Significance

The success of the AHR mechanism carries several important implications for the design and application of MoE models.

- **Improved Interpretability**: The emergence of highly specialized experts in the final layers offers a promising avenue for model interpretability. With such clear functional differentiation, it may become possible to analyze and even assign semantic labels to individual experts (e.g., the "Sports Terminology Expert" or the "Scientific Discourse Expert"). This could provide unprecedented insight into how large models represent and process complex information.

- **Potential for Enhanced Transfer Learning and Fine-tuning**: A model architecture with a generalist base and a specialized top may be inherently better suited for transfer learning. For a new downstream task, one could envision freezing the weights of the generalist early layers and fine-tuning only the specialized final-layer experts. This could lead to more parameter-efficient and robust knowledge transfer, as the foundational feature representations would be preserved.

- **A New Design Principle for MoE Routers**: AHR introduces the principle of *contextual hierarchical routing*. This shifts the paradigm of MoE router design away from the prevailing focus on stateless, token-only routing decisions. It opens up a new and rich design space where the router is not just a selector but an orchestrator, guiding the flow of information through the network in a stateful and context-aware manner.

## 6.3 Limitations and Future Directions

While the results are promising, this work has several limitations that open avenues for future research.

- **Scope of Evaluation**: The experiments were conducted on a single, although standard, text classification task. To establish the generalizability of AHR, future work must validate its effectiveness across a broader range of tasks, such as language generation, summarization, and code completion, as well as across different data modalities such as vision.

- **Scalability**: The current experiments utilize a small 4-layer model with 8 experts per MoE layer. This small-scale success offers no guaranty of performance or stability on larger scales. The challenges of load balancing and specialization become more acute as expert counts and model depths increase. Therefore, a crucial and mandatory test is to investigate if AHR's specialization benefits persist (or even exist) in genuinely large models (e.g., with 64, 128, or more experts) and within significantly deeper architectures.

- **Architectural Variations**: The AHR implementation presented here uses a simple linear projection to generate the astrocytic bias. Future research could explore more complex, non-linear functions (e.g., small MLPs) or even attention mechanisms to create more sophisticated and powerful global-local routing dependencies.

- **Interaction with Advanced Load Balancing**: This work employed a standard auxiliary loss for load balancing. An interesting direction for future study would be to investigate the interplay between AHR and more advanced techniques such as loss-free balance (Wang et al., 2024). It is plausible that AHR's inherent structure, which naturally channels tokens down specific pathways, might reduce the need for strong, explicit balancing constraints, potentially leading to a more synergistic and performant system.

- **Adaptation to Decoder-Only Architectures**: The current study utilizes a BERT-style encoder, where the `[CLS]` token provides a natural, bidirectional, sequence-level context vector. This approach is not directly transferable to modern auto-regressive, decoder-only LLMs, as their causal attention mechanism means the beginning-of-sequence (BOS) token lacks global information. A critical future direction is to adapt AHR for these architectures. We hypothesize that the "astrocytic context"

can be derived from the user's *prompt*. For example, one could compute a single context vector by mean-pooling the hidden states of all prompt tokens. This static "prompt context" vector would then be used to additively bias the routing decisions for all subsequently generated tokens, retaining the core AHR principle of using a stable global signal to modulate local routing. This proposed adaptation is entirely theoretical and remains a critical, unverified hypothesis that must be tested from scratch.

## 7 Conclusion

This paper addressed the persistent challenge of achieving genuine expert specialization in Mixture-of-Experts models. The standard approach often leads to a "specialization paradox," where mechanisms for load balancing inhibit the very functional differentiation they are meant to enable. To resolve this, we proposed Astrocyte-Hierarchical Routing (AHR), a novel routing mechanism inspired by the modulatory role of astrocytes in the brain. AHR introduces a contextual, hierarchical dependency into the routing process, conditioning local, token-level expert selection on a global, sequence-level context vector derived from the [CLS] token. Through extensive experiments on a text classification benchmark, this study demonstrated that AHR successfully cultivates a developmental trajectory for expert function. It fosters generalist behavior in early layers and promotes the emergence of highly specialized experts in the final, most abstract layers of the network. Crucially, this significant enhancement in specialization is achieved without any statistically significant degradation in primary task accuracy compared to strong, well-tuned baselines. This work not only provides a practical and effective method for improving the functional properties of MoE models but also introduces a new design principle for routing mechanisms, paving the way for the development of more sophisticated, context-aware, and ultimately more powerful conditional computation architectures.

### Broader Impact Statement

This work aims to improve Mixture-of-Experts (MoE) models to reduce the high energy and financial costs of large-scale AI, potentially democratizing access. Furthermore, our method (AHR) creates specialized experts, which may improve AI safety and interpretability. This could simplify auditing for fairness, identifying bias, and debugging failures. However, risks exist. First, more efficient models are easier to misuse for malicious purposes, such as disinformation or surveillance. Second, our specialization mechanism could amplify bias. An expert might hyper-specialize in a harmful stereotype, making the model's outputs more systematically biased than a dense model. Finally, this specialization may reduce robustness, making models brittle against novel or adversarial inputs. These trade-offs require further investigation.

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
