# OpenReview forum: "Astrocyte-Inspired Hierarchical Routing for Enhanced Expert Specialization in Mixture-of-Experts Models"
_TMLR — Rejected by TMLR_

### Review · Reviewer_KsvZ · 2025-11-23

**Summary Of Contributions:**

The paper proposes Astrocyte-Hierarchical Routing (AHR), a method that modulates local token-level routing decisions in Mixture-of-Experts (MoE) models using a global context signal. In the implemented encoder-based architecture, the authors project the [CLS] token representation into a bias vector that is added to the router's logits. The authors evaluate this method on a small 4-layer Transformer encoder using the AG News text classification dataset. The reported results indicate that AHR achieves task accuracy comparable to standard Softmax-MoE and Dense baselines while resulting in higher variance in expert utilization in the final layer.

**Strengths**
- The proposed additive global bias mechanism is architecturally simple and introduces negligible parameter overhead.
- The visualization of layer-wise expert utilization patterns effectively illustrates the shift in expert behavior across the network depth.

**Audience:**

No

**Audience Explanation:**

The findings are restricted to a specific, small-scale legacy architecture on a toy task. Given that the community has moved toward large-scale decoder-only MoEs, a routing mechanism that yields no significant performance gain and has not been tested on relevant architectures or against current baselines is unlikely to be of interest.

**Broader Impact Concerns:**

I have no concerns regarding the broader impact of this work that would require a specific statement.

**Claims And Evidence:**

No

**Claims Explanation:**

- The experimental setting consists of a 4-layer, 8-expert BERT encoder trained on a single classification task. MoE architectures are notoriously sensitive to scale, and issues such as representation collapse or routing instability often manifest differently in large-scale models compared to tiny ones. Validating a routing strategy on such a microscopic scale without standard large-scale benchmarks is insufficient to demonstrate its effectiveness.

- The method relies on the [CLS] token, which makes it inherently incompatible with the dominant MoE architectures used in current research, which are almost exclusively Decoder-only models like Mixtral, Qwen-MoE, or DeepSeek-MoE. The authors acknowledge this limitation but do not address it experimentally.

- The baseline comparisons are insufficient. The paper compares AHR only against standard Top-K Softmax and a Dense baseline, ignoring relevant recent literature that specifically addresses expert specialization. For instance, the authors do not compare against or discuss recent relevant works such as Advancing MoE Efficiency: A Collaboration-Constrained Routing (C2R) Strategy for Better Expert Parallelism Design (Zhang et al., 2025) or Advancing Expert Specialization for Better MoE (Guo et al., 2025).

- The paper claims enhanced specialization as a contribution but fails to demonstrate the practical utility of this specialization. The task accuracy is statistically indistinguishable from the baseline. If specialization does not lead to improved accuracy or efficiency, its value is unclear. The paper lacks quantitative metrics typically used in specialization literature to justify why this specific routing behavior is preferable to the baseline.

**Requested Changes:**

- The authors should implement and test AHR on a Decoder-only architecture to prove the method works with causal masking, rather than relying on [CLS] token encoders.
- The experiments must be moved beyond the 4-layer toy model to a standard base-sized model (e.g., 3B) or larger to show the method holds up under realistic conditions.
- The authors need to include comparisons with modern routing strategies relevant to specialization and efficiency, such as Zhang et al. (2025) and Guo et al. (2025), rather than just standard Top-K.
- The authors should provide evidence that the increased specialization score translates to a tangible benefit, such as improved efficiency or performance trade-offs, given that accuracy remains unchanged.

---

### Review · Reviewer_r2oy · 2025-11-27

**Summary Of Contributions:**

This paper proposes Astrocyte-Hierarchical Routing (AHR), a bio-inspired expert selection mechanism for Mixture-of-Experts (MoE) models that encourages expert specialization without degrading task performance. In encoder-based MoE models, AHR introduces a global, sequence-level bias (derived from the [CLS] token) into token-level routing logits, so that routing decisions depend not only on local token representations but also on global context. Experiments on AG News text classification with BERT-based MoE models show that AHR matches the accuracy of both softmax-gated MoE and dense baselines, while yielding substantially higher expert specialization in the final layer.

**Audience:**

Yes

**Audience Explanation:**

The paper introduces a new routing mechanism that improves expert modularity in MoE models by leveraging global information with a simple modification, which is relevant to researchers working on interpretability. The observation that AHR reduces expert specialization in the first layer while substantially increasing it in the final layer is particularly interesting. In addition, the bio-inspired design, drawing on astrocyte-like modulation, will likely appeal to parts of the community interested in neuroscience-inspired architectures.

**Broader Impact Concerns:**

All the concerns and ethical implications are discussed in the Broader Impact Statement section.

**Claims And Evidence:**

Yes

**Claims Explanation:**

The paper evaluates several BERT-based MoE variants on a multi-class text classification task, comparing AHR to softmax-gated MoE baselines and dense models. Results over 5 random seeds are reported, and statistical significance is assessed using t-tests. The authors show that AHR's accuracy is statistically indistinguishable from the dense and softmax-based baselines. At the same time, their specialization metric (based on the variance in expert utilization across topics) clearly shows higher expert specialization in the final layer, while lower specialization in the first layer. This supports the claimed "developmental trajectory" from generalist to specialist experts.

**Requested Changes:**

Critical
- Please expand the discussion of how the AHR design could be adapted to decoder-only architectures, which are dominant in modern LLMs. The current discussion in the limitations section is not very concrete.
- Please include at least some evaluation on a non-classification task. For classification, the [CLS] representation is explicitly trained to capture information about sentence types, so the effectiveness of [CLS]-based global bias in expert specialization is somewhat expected; it would be valuable to know how AHR behaves in other settings.
- Please provide more details about the energy loss used in Section 4.2.
- Please clarify the meaning of the LB and EN components in the model configuration names (e.g., "LB0.01", "EN0.01").
- In the description of Figure 5 (page 9), the reported numbers, such as 59.6% and 52.7% were not immediately obvious to me from the figure. Please clarify exactly how these percentages are computed and how they correspond to the visualized heatmap values.

Simply to strengthen the work
- For the astrocyte-meanpool setting, please explain why you chose to multiply $L_{token}$ by $w_{global}$, rather than using an addition, as in Equation (7). A short rationale or ablation (if available) would clarify whether the benefit comes from the source of the context signal, from additive vs multiplicative modulation, or both.
- Figure 1: Some of the text overlaps slightly; improving the layout or font size would make the figure easier to read.
- Figures 2 and 3: If space permits, it would be helpful to report specialization scores for intermediate layers as well.
- For all figures, it would improve readability to use consistent colors for the same model configurations across plots.
- It would be interesting to add a brief discussion of why the softmax baselines exhibit higher specialization scores in the first layer than AHR. This could provide more intuition about how standard load balancing and routing differ from the hierarchical bias introduced by AHR.
- In Figures 4 and 5, some experts appear to be almost unused (e.g., Layer 2 E1 in Figure 4 and Layer 1 E0 in Figure 5). Please comment on whether this is expected given the load-balancing and energy losses.
- It would be interesting to discuss whether AHR could be extended so that every layer achieves meaningful expert specialization, rather than primarily the final layer. One potential benefit would be more predictable expert usage across depth, which could be especially useful for offloading-based systems where knowing in advance which experts are likely to be activated is important.

---

### Review · Reviewer_vcc2 · 2025-12-29

**Summary Of Contributions:**

This work proposes a new method for MoE expert selection built on top of the classic token-based approach. The proposed method introduces an additive context-based bias on top of the token-based logits. The context-based bias is computed using the [CLS] token’s representation at the beginning of the sequence, which contains the information of the entire sequence due to self-attention operations in the encoder. The work only works for the encoder model and performed experiment on the AG News benchmark dataset with one single model configuration. The experiment results show that the proposed approach achieves similar classification accuracy compared to the classic softmax MoE approach while increasing the specialization of the experts in later layers of the model.

**Additional Comments:**

**Additional questions**:

What’s the scale ratio between L_token and L_global in equation 7? Will the context-based L_global dominate the expert selection in some cases?

Is there a direct comparison of the proposed method with other similar approaches listed in the related works on the same dataset?

The dataset and task used in the work only have 4 classes, making it very easy to determine the context. How will the proposed method perform and be evaluated if the context is much richer or even without ground truth labels (for example, in some decoder tasks)?

**Audience:**

Yes

**Audience Explanation:**

**Strength**:

**Very simple and efficient approach with high effectiveness on expert specialization**: The proposed approach only requires a minimal computation increase by computing the linear transformation of a single token. However, with such a simple addition, the encoder expert routing activity is effectively changed to the desired behavior.

**Potential to work on any encoder models without major change**: Since the proposed method directly works on the first token of the sequence, it can be used on any encoder model where the first token (or any non-text token) contains the full context information of the sequence.

**Broader Impact Concerns:**

There are no broader impact concerns.

**Claims And Evidence:**

No

**Claims Explanation:**

**Weakness**:

**Weak link to astrocyte or neuronal network inspiration**: Although the work claims to be astrocyte-inspired, the link to the actual mechanism of astrocytes is very weak. Indeed, the tripartite synapse enables the astrocyte to influence individual synapses using global information. However, this kind of influence occurs on a very slow timescale compared to synaptic activities and is mainly used to maintain the stability and dynamics of the neural system. What is proposed in this work operates at a much higher level of abstraction, making the proposed approach more related to conditional computing rather than the synaptic or network dynamic level. Claiming the work is astrocyte-inspired is a bit of a stretch.

**Limited experiments cannot support the claim with high confidence**: The final claims of the work are built on experiments based on only one task, one dataset, and one model architecture. Therefore, it is impossible to justify that the findings are generalizable. The authors need to show, at a minimum, that the claims can be generalized to a wider range of encoder-based models and tasks.

**Experimental design fails to demonstrate the benefits of expert specialization**: The experiments in this work only compare accuracy and expert specialization across different approaches. Since the results show that the proposed method yields similar accuracy to the classic MoE approach, the advantage of increasing expert specialization is not clearly demonstrated. While the experiments show a new method that can alter expert routing behavior without hurting performance, they do not explain why this change is beneficial. The authors need to design additional experiments to show a clear advantage of their approach.

**Requested Changes:**

**Potential changes that can improve the work**:

1. Remove the claim on bio inspiration and address the proposed work from the conditional and context-aware computing perspective.

2. Add a dataset for encoder models with a more complex context (>> 4 classes). Perform experiments with different sizes and depths of the encoder network and show if the effects are observed.

3. Design a special experiment to show the advantage of expert specialization. For example, perform an experiment on model interpretation or model generalization.

---

### Comment · Action_Editor_Hhu6 · 2025-12-29
**All reviews are submitted - discussion period starts**

Dear Authors,

All reviews have now been posted, and the discussion period is officially open.

The Reviewers raised several critical and recurring concerns that need to be addressed in your response and manuscript revision. To mention the most salient ones: the framing using astrocyte as inspiration has been criticised, with reviewers suggesting the work be re-contextualised as a form of conditional computing; the experimental scope is considered limited; and finally, the practical benefits of the observed increase in expert specialisation appear unclear.

Please use this discussion period to clarify these doubts and improve the manuscript.

Looking forward to your response.

Regards, AE

---

> ### Author Response · Authors · 2026-01-08
> **Author Response: Hierarchical Contextual Routing (HCR) for Enhanced Expert Modularity and Pruning in Decoder-Only Upcycled Mixture-of-Experts**
>
> Dear Action Editor and Reviewers,
>
> We appreciate the insightful critiques provided during the review process. In response, we have executed a fundamental pivot: reframing the work from a biological metaphor to a Proof-of-Concept for a conditional computing framework and validating its utility on decoder-only architectures (GPT-2). These updates also demonstrate the tangible benefits of expert modularity for model compression.
>
> I. Conceptual Re-contextualization (AE, vcc2, r2oy):
>
> We have moved away from the "astrocyte" terminology to clarify the technical contribution):
> * New Framework: We now define the work as **Hierarchical Contextual Routing (HCR)**.
> * Manuscript Changes: The title, abstract, and introduction will be rewritten to focus on conditional computing and expert modularity. Biological references will be restricted to a brief inspiration note.
>
> II. Validation on Decoder-Only Architectures (AE, KsvZ, r2oy):
>
> To address concerns regarding decoder compatibility, we implemented HCR within a GPT-2 (4-layer, 8-expert) architecture using four distinct routing configurations to isolate the effects of context.
> 1. Adaptation and Ablation Configurations:
> * Softmax Baseline: Standard token-level routing without contextual bias.
> * HCR-Prompt: Constant global bias using a mean-pooled representation of the initial tokens.
> * HCR-Recent: Constant local bias using a semantic signal from the $K=5$ most recent hidden states.
> * HCR-Progressive: A layer-adaptive blending schedule that transitions from local to global context:
> $$\lambda = \frac{l}{L-1}$$
> $$C_{final} = (1-\lambda)C_{recent} + \lambda C_{prompt}$$
> This creates a trajectory where early layers act as generalists and later layers act as context-driven specialists.
>
> 2. Experimental Results (N=5 seeds)
> * Accuracy Parity: All variants maintained performance parity ($p > 0.05$). HCR-Progressive ($0.9494 \pm 0.0016$) vs. Softmax Baseline ($0.9499 \pm 0.0008$) yielded $p=0.328$.
> * Specialization Hierarchy: HCR-Progressive successfully induced the intended trajectory, showing a 28.8% increase in final-layer specialization ($p=0.008$) and a 43.0% decrease in first-layer specialization ($p=0.001$) relative to Softmax.
> * Contextual Modularity: The HCR-Recent ablation showed significantly lower specialization in both first ($0.0109$) and last ($0.0619$) layers compared to Softmax ($0.0147$ and $0.0762$), confirming that fixed local context alone is insufficient for high-level expert specialization.
>
> III. Tangible Benefits of Expert Modularity: Pruning and Efficiency (vcc2, KsvZ):
>
> To demonstrate practical utility, we conducted Pruning Capacity Analysis, measuring the maximum expert removal possible while maintaining accuracy degradation $<0.5\%$).
> * Hypothesis: HCR-recent have an increased pruning capacity (following more generalists experts)
> * Results: The HCR variant achieved 74.2% pruning capacity, an 8% relative improvement over the Softmax baseline (68.7%).
> * Impact: By inducing contextual modularity (with recent local-context window), HCR creates more compressible, modular expert pathways.
>
> IV. Consolidated Revisions & Reviewer Concerns
>
> We have merged the following technical updates and responses into the revised manuscript:
> * Addressing the "Energy Loss" Confound (r2oy): We will clarify that while $L_{energy}$ (misleading name for L2 penalty) ensures stability, the specialization is driven by the contextual bias. HCR-Recent and HCR-Progressive yield distinct modularity patterns despite identical loss coefficients ($\alpha=0.01$).
> * Formalizing Notation (KsvZ): We will define the scaling ratio $\gamma$ between token and global signals:
>   $$R(x) = \text{Top-K}(W_g x + \gamma \cdot C_{final})$$
>   This ensures the global bias modulates rather than dominates the routing. We will add a derivation of the specialization percentage metric in the Appendix.
> * Comparison to Related Work (KsvZ): A new section will compare HCR to Collaboration-Constrained and Orthogonal routing. We position HCR as a complementary input-side bias that requires negligible parameter overhead.
> * Scale and Scope (KsvZ, vcc2): We will reframe the paper as a Proof-of-Concept for Contextual Routing Design. We acknowledge the 4-layer scale and will provide a roadmap for MoE "Upcycling" (applying HCR to pre-trained models LLM with >3B parameters) as a pathway for large-scale deployment on more complex datasets (with higher cardinality than AG news), providing a more controlled experimental ground than MoE models trained from scratch.
> 7. Discussion of Limitations
> * Visual Clarity: Figures 1-3 will be revised to remove text overlap and include intermediate layer specialization scores to better visualize the "Generalist-to-Specialist" trajectory.
>
> We believe these revisions—specifically the transition to HCR and the new decoder-based pruning results—directly resolve the core concerns regarding compatibility and utility.
>
> Sincerely,

---

### Decision · Action_Editor_Hhu6 · 2026-02-01

**Recommendation:** Reject

**Audience:**

Yes

**Audience Explanation:**

Yes, some of the points are interesting and if confirmed would benefit the people interested in modularity and interpretability. However, at the moment, the results are incomplete because of the scale issue discussed above. Additional validations are needed to strongly support the claims,

**Claims And Evidence:**

No

**Claims Explanation:**

The manuscript has significantly improved over the discussion period, with the authors addressing concerns regarding the decoder compatibility, the practical advantages of expert specialisation, and the absence of relevant baselines. The experiments on a decoder-only architecture and the inclusion of pruning capacity analysis provide evidence that higher specialisation facilitates compression. Additionally, they moved away from the "astrocyte" metaphor which was unanimously criticised.

However, the issue regarding scalability remains standing. The experimental validation relies on a 4-layer architecture, which the authors claim to serve as a "proof-of-concept." However, this represents a regime where MoE challenges are moderate. Given that MoE architectures are sensitive to scale and often exhibit routing instabilities or representation collapse only in larger settings, validating a strategy on such a small scale is insufficient to generalise its effectiveness. Since the reported results demonstrate only accuracy parity without significant performance gains, the claims of practical utility cannot be substantiated without larger-scale experiments that truly stress the routing mechanism.

**Resubmission Of Major Revision:**

The authors may consider submitting a major revision at a later time.